# Mediating Similarity: An Information-Theoretic Principle of Reference Behavior

**DOI:** 10.3390/e27111124

**Published:** 2025-10-31

**Authors:** Qun Zhao, Menghui Yang, Guojian Xian, Jieying Bi, Tan Sun

**Affiliations:** 1Agricultural Information Institute, Chinese Academy of Agricultural Sciences, Beijing 100012, China; zhaoqun@caas.cn (Q.Z.); xianguojian@caas.cn (G.X.); bijieying@caas.cn (J.B.); 2School of Information Resource Management, Renmin University of China, Beijing 100872, China; yangmenghui@ruc.edu.cn; 3Key Laboratory of Data Engineering and Knowledge Engineering, Ministry of Education, Beijing 100872, China

**Keywords:** reference behavior, information theory, Kullback-Leibler divergence, scientometrics, mediating similarity

## Abstract

While information theory is widely used to quantify knowledge combinations, the fundamental principles guiding reference selection in science remain largely unexplored. This study analyzes a large-scale journal citation network to introduce and empirically validate a principle we term “Mediating Similarity”. We posit that a journal’s reference list acts as a strategic cognitive bridge, creating a more efficient informational path from its specific research identity to the broader scientific landscape. Using information-theoretic measures and computational experiments, we tested this principle and its underlying mechanisms. Our findings provide robust, multi-level evidence. First, we confirm the universality of the principle, showing that the mediated path through references is consistently more efficient than the direct path for thousands of journals. Second, perturbation experiments reveal a dual mechanism guiding reference selection: real-world reference portfolios are not merely collections of relevant works, but are synergistically optimal combinations that vastly outperform randomly assembled alternatives. This global optimization, however, operates as a robust “satisficing” strategy, balancing the search for an ideal cognitive path with the practical constraints of scientific discovery. Collectively, these findings reframe reference behavior as a strategic process of navigating a cognitive energy landscape, where journals selectively curate references to enhance their integrative capacity and innovative potential.

## 1. Introduction

Information theory, originally rooted in thermodynamics, was later applied to measure uncertainty in information systems. In scientometrics, it is utilized to quantify the diversity and heterogeneity of knowledge combinations. The field of scientometrics, which emerged in the mid-20th century with foundational works by Derek J. de Solla Price’s Little Science, Big Science [1] and Eugene Garfield’s pioneering efforts in citation analysis and the Science Citation Index [2,3], aims to study scientific practices and their effects through quantitative data. Early applications of information entropy in scientometrics, such as in [4], demonstrated its utility in measuring research institutions’ involvement in R&D and assessing the breadth and depth of national technology strategies. Since then, information entropy has become an indispensable tool for understanding the complex dynamics of scientific systems. Various entropy concepts and related information-theoretic measures have been extensively applied across diverse aspects of the science of science. Shannon entropy, as a fundamental measure of uncertainty [5], and Kullback–Leibler (KL) divergence [6], which quantifies the difference between two probability distributions, are particularly prominent.

Innovation emerges through the combinatorial process of exploring and integrating existing concepts. From a social network perspective, this process depends on the underlying structure of connections. Uzzi and Spiro [7] investigated creative idea genesis and found that groundbreaking achievements in creative industries arise from teams that bridge disparate knowledge clusters, characteristic of "small-world" networks. These networks combine dense local connectivity with long-range ties, facilitating both the development of existing ideas and the formation of novel combinations. Uzzi et al. [8] analyzed 17.9 million papers and found that while most scientific work builds on conventional knowledge combinations, high-impact studies are characterized by familiar conceptual foundations punctuated by novel, atypical combinations. This suggests breakthrough discoveries require balancing reinforced pathways of established knowledge with exploratory jumps to adjacent, less-traveled parts of the conceptual network. Iacopini et al. [9] model innovation as an “edge-reinforced random walk” where researchers’ attention moves between linked concepts. Innovation occurs when a new concept is visited for the first time, capturing the “adjacent possible” principle where discoveries emerge in the conceptual neighborhood of known ideas.

The evolution of scientific fields through novel idea combinations can be measured using information-theoretic tools like Kullback–Leibler (KL) divergence. Chen [10] applied KL divergence to analyze co-citation network dynamics, measuring year-over-year changes in betweenness centrality distributions for scientific articles. Higher KL divergence values, indicating substantial structural network changes, predicted which articles would later achieve high citation impact. This demonstrates KL divergence as an early indicator of transformative research. Similarly, Leydesdorff and de Nooy [11] utilized entropy statistics to map scientific “hot spots” by analyzing journal citation dynamics, identifying critical transitions and unexpected changes at both journal and citation levels. This approach is not limited to citation networks; modern analyses of dynamic networks, such as those in online political or public opinion discourse, similarly rely on information theory to quantify complexity, uncertainty, and influence [12,13].

Changes in reference patterns predict innovation, but what factors guide this selection? Less is known about the principles that govern how science navigates the vast knowledge landscape to select references. Recent work frames social power not just in terms of resources, but as the ability to efficiently process information, which confers a competitive advantage [14]. Do journals similarly engage in a strategic selection of references to enhance their information-processing capacity, thereby boosting their potential for innovation? This paper introduces and empirically validates a principle we term “Mediating Similarity”.

We propose that a journal’s reference list acts as an informational intermediary, or “cognitive bridge,” between its own scientific position and the broader knowledge environment. To formalize this, we define three key probability distributions: (1) Journal’s Citation Distribution, which represents its scientific identity based on how journal cites; (2) Journal’s Reference Distribution, representing the collective knowledge based on how focal journal’s cited journals cite; and (3) the Overall Scientific Content Distribution, representing the entire knowledge environment based on how all journals cite. Our principle posits that the informational “distance” from a journal’s identity to the global landscape is shorter when routed through its references than when measured directly. In information-theoretic terms, where this distance is quantified by Kullback–Leibler (KL) divergence, “Mediating Similarity” is captured by the inequality: KL(CitationDist.‖ReferenceDist.)
+KL(ReferenceDist.‖OverallDist.)
<KL(CitationDist.‖OverallDist.)

Our analysis of a comprehensive journal citation network provides strong empirical support for this principle through three key findings. First, we establish the universality of the mediating effect: for all 19,129 journals tested, the mediated cognitive path through their actual references is informationally shorter than the direct path. Second, our results reveal that this efficiency is the product of a powerful global optimization. Through large-scale perturbation experiments—initially on 500 top journals and then rigorously validated on a random sample of 500 journals—we demonstrate that actual reference portfolios are synergistically optimized combinations. On average, a real portfolio’s ’cognitive energy’ ranks in the bottom 1.34% when compared to thousands of randomly assembled yet highly relevant alternatives. Third, this global optimization operates as a robust ’satisficing’ strategy. In local perturbation tests where 10% of references were swapped, the actual portfolios still outperformed the vast majority of slightly altered variations (ranking in the 19th percentile on average). This suggests the selection process is a boundedly rational search that balances the drive for an ideal cognitive path with the practical constraints of scientific discovery.

## 2. Related Works

The view that scientific innovation is a combinatorial process has deep philosophical and economic roots. Physicist Max Planck argued that science is fundamentally a single, unbroken chain of knowledge—from physics through chemistry and biology to anthropology and the social sciences—whose apparent divisions are merely a product of limited human cognition [15]. Kuhn [16] further developed a holistic view, challenging the idea of simple, linear progress. He proposed that science is not an accumulation of isolated facts but a complex “constellation” of validated observations, conceptual frameworks, and procedural techniques. For Kuhn, true innovation involves rearranging this constellation or proposing a new configuration entirely—a process captured in his concept of the “paradigm shift.” This combinatorial perspective is echoed in economics by Joseph Schumpeter, who defined innovation as the “new combination” of production factors to create novel products, methods, or markets [17]. If the process of innovation is one of constantly changing combinations, the challenge then becomes how to describe and quantify this fusion and transmutation. Here, information theory, with its capacity to measure complexity and uncertainty in complex systems, offers a powerful toolkit [5].

The application of entropy to measure the diversity of knowledge combinations is a central theme in the literature. Innovation is often conceptualized as a combinatorial process where novel ideas arise from new mixtures of existing knowledge [8]. Entropy-based metrics have proven effective at quantifying this diversity in several ways, such as measuring citation diversity to assess interdisciplinarity and scientific disruptiveness [18,19,20]. A critical application of these methods is in scientific trend identification and prediction. Chen [10] demonstrated that KL divergence can be used to quantify the year-over-year change in the betweenness centrality distribution of articles in a co-citation network; a higher divergence value, indicating significant structural variation, was found to be a robust predictor of future citation impact. Similarly, Leydesdorff and de Nooy [11] utilized KL divergence to map scientific “hot spots” by identifying critical, non-linear transitions in journal-journal citation networks. These studies show that changes in citation patterns, viewed as combinations of reference knowledge, can predict innovation.

This raises a fundamental question: if certain patterns predict success, what underlying process drives the formation of these advantageous patterns? The investigation into authors’ motivations for citing is a complex field, marked by a long-standing theoretical debate between two primary viewpoints: the normative theory and the social constructivist theory [21]. The normative theory, rooted in the work of [22], posits that citation is a moral and intellectual obligation to acknowledge the influence of prior work, functioning as a “pellet of peer recognition” within the scientific reward system. In contrast, the social constructivist view, championed by sociologists like [23], argues that citation is primarily a tool of persuasion and rhetoric. From this perspective, authors strategically select references to bolster their claims, appeal to authorities, and convince their audience.

Empirical studies have shown that real-world citation behavior is a blend of both. In a comprehensive meta-synthesis, Lyu et al. [24] categorized citing motivations into “scientific” and “tactical.” Scientific motivations align with the normative view and serve direct rhetorical functions, such as providing Background, identifying a research Gap, establishing a theoretical Basis, or offering Evidence. Tactical motivations, aligning with the social constructivist view, are more socially or benefit-oriented, such as adhering to a Subjective Norm (e.g., citing a reviewer’s work), Advertising one’s own expertise, or Profit-seeking (e.g., increasing the chances of publication). Case and Higgins [25] surveyed authors in the field of communication, provides concrete evidence for this blend; they found that authors cite works because they are “concept markers” (a normative function) but also to “promote the authority of one’s own work” (a persuasive, constructivist function).

The complexity of citation behavior is further deepened by disciplinary and contextual factors. Nishikawa [26] revealed in a citation context analysis that the patterns of knowledge flow between natural sciences (NS) and social sciences/humanities (SSH) are non-intuitive; for instance, NS papers frequently cite SSH papers in their Methodology sections, indicating a methodological contribution from SSH to NS, a pattern contrary to common assumptions. Even within a single category of motivation, behavior is nuanced. Bordignon [27] highlights the “paradox of the critical citation,” noting that while criticism is fundamental to scientific progress, direct negative citations are rare. Instead, authors often employ more subtle strategies like comparison or questioning, suggesting that even oppositional citation is a strategic and carefully managed act.

However, investigating these motivations is fraught with methodological challenges. Content analysis by third-party coders risks being subjective, while direct author surveys, as used by [25], suffer from potential recall bias and social desirability bias, where authors may be reluctant to admit to tactical motivations [21,24]. The rise of automated citation function classification using deep learning has introduced a new dynamic: the need for computationally feasible annotation schemes often leads to a simplification of the rich, theoretical taxonomies developed in bibliometrics [28].

While this large body of work provides detailed taxonomies of what motivates citations and how these motivations vary across contexts, it has yet to produce a unifying principle that explains why the scientific system organizes itself through these specific behavioral patterns. The existing models describe the pieces of the puzzle—persuasion, intellectual debt, social norms—but do not explain the emergent, system-level logic that governs the selection of a reference portfolio as a whole. Albarracin et al. [14] offers a compelling perspective by reframing social power as the ability to efficiently process information. When applied to the scientific ecosystem, this suggests that a journal’s innovative capacity may be deeply tied to how strategically it processes the vast information landscape through its citation choices. This study addresses this gap by introducing and empirically validating the phenomenon of “Mediating Similarity.” We propose that a journal’s reference list acts as a cognitive bridge, and that the selection process is guided by a principle of optimizing this cognitive path. By framing reference behavior as a strategic process of minimizing “cognitive energy,” this work aims to provide a fundamental, information-theoretic mechanism that underlies the diverse motivations and complex patterns observed in previous research.

## 3. Methods

### 3.1. Data

The dataset was collected in February 2021 from the OpenAlex database [29]. As the successor to the Microsoft Academic Graph [30], OpenAlex indexes a comprehensive range of academic publications, making it a suitable choice for this analysis in terms of data scale and subject coverage. The dataset contains metadata for approximately 200 million academic articles. Using the citation information provided in these articles, a journal citation network was constructed, represented as a weighted directed graph with 99,152 nodes (journals) and 131,098,937 edges (citation relationships between journals).

For the primary analysis, this study selected journals from the 2021 SCImago Journal Rank (SJR) list [31]. This list was chosen for two primary reasons. First, its public accessibility ensures the transparency and reproducibility of our selection process. Second, while numerous journal evaluation metrics exist (e.g., Journal Impact Factor, CiteScore), they are generally highly correlated. Therefore, using the widely recognized SJR list provides a representative and robust sample of reputable journals without introducing significant selection bias compared to other established indicators. The SJR list includes 27,339 journals, of which 19,129 were found in the OpenAlex database. These 19,129 journals were used to construct the citation network for our experiments.

### 3.2. Non-Parametric Statistical Methods

This paper employs a non-parametric statistical method to estimate a population distribution through a weighted combination of individual sample distributions. The weighting is designed to preserve the relative importance of each sample. Given a set *D* with *n* samples, $D=X_{1},X_{2},\ldots ,X_{n}$, each sample $X_{i}$ is first represented as a vector $V_{i}$. Each vector can be seen as a distribution with a magnitude, $v_{i}/\left|\right|v_{i}\left|\right|$. The magnitude of each sample distribution vector, $\left|\right|v_{i}\left|\right|$, is used as its weight in the estimation of the overall population distribution. This provides a natural way to represent the relative importance of each sample. The estimated population distribution, $\hat{p}\left(x\right)$, is obtained by summing all individual sample distribution vectors: $\hat{p}\left(x\right)=\left(\sum_{i}v_{i}\right)/\left|\right|\sum_{i}v_{i}\left|\right|$. This method is closely related to Kernel Density Estimation (KDE) [32] and Gaussian Mixture Models (GMM) [33]. Like KDE, it estimates the overall distribution by superimposing local distributions. However, unlike KDE which uses a uniform kernel function, our method allows each sample to have a unique distribution form and weight. Similar to GMM, our method uses a weighted sum of multiple distributions to represent a complex distribution. However, unlike GMM’s fixed number of components and explicit weights, the number of components in our method equals the number of samples, and the weights are implicitly defined by the vector magnitudes.

To convert the raw citation count vectors into probability distributions suitable for KL-divergence, we normalize each vector by dividing each of its components by the sum of all its components. This ensures the elements of each resulting distribution sum to 1. This paper refers to $\frac{V_{i}}{\sum_{k}v_{i,k}}$ as the Journal’s Citation Distribution. The sum of the degree vectors of all journals cited by a given journal is treated as a distribution estimated by the aforementioned non-parametric method. This paper refers to this as the Journal’s Reference Distribution. Existing research has shown that the complexity of this distribution can be measured to assess a journal’s diversity, a method known as Entropy of Degree Vectors Sum (EDVS) [34].

The Journal’s Citation Distribution ($d_{1}$) can be seen as a representation of an individual journal’s research content. The Journal’s Reference Distribution ($d_{2}$) represents the content of a journal’s references. The sum of the degree vectors of all journals in the citation network is considered to represent the content of the knowledge environment. This distribution transcends the scope of individual journals and disciplines, representing the overall attention distribution in science. The Overall Scientific Content Distribution ($d_{3}$) is thus a representation of the content of the knowledge environment. To address the issue of zero probabilities in the distributions, which would lead to undefined Kullback–Leibler divergence values, we employed an additive smoothing technique. Before normalizing the raw citation count vectors to create the probability distributions ($d_{1}$, $d_{2}$, and $d_{3}$), we added a small smoothing parameter α to every count in each vector. For this study, we set α to a small constant, 1 × $10^{-9}$. We selected this extremely small value to ensure that the smoothing has a negligible impact on the results, while still achieving numerical stability. The raw data for citation counts ranges from a minimum of 1 to potentially hundreds of thousands. This procedure ensures that all probabilities are non-zero, guaranteeing the numerical stability of all KL divergence calculations while having a minimal impact on the original distribution’s structure. $d_{1}$, $d_{2}$ and $d_{3}$ are shown as Equation (Equation 1). We use a simple network as a toy model to demonstrate the calculation of three distributions in Figure 1.(1)$$\begin{array}{cc} d_{1}\left(i\right) & =\frac{V_{i}+\alpha }{\left(\sum_{k}v_{i,k}+\alpha \right)}\\ d_{2}\left(i\right) & =\frac{\sum_{j\in {\mathrm{neighbors}}_{i}}V_{j}+\alpha }{\left(\sum_{j\in {\mathrm{neighbors}}_{i}}\sum_{k}v_{j,k}+\alpha \right)}\\ d_{3} & =\frac{\sum_{i\in \mathrm{all}}V_{i}+\alpha }{\left(\sum_{i\in \mathrm{all}}\sum_{k}v_{i,k}+\alpha \right)} \end{array}$$

Kullback–Leibler (KL) divergence, also known as relative entropy, is an asymmetric measure of the difference between two probability distributions, *P* and *Q*. It quantifies the extra average number of bits required to encode samples from a distribution *P* when using a code based on a distribution *Q*, representing uncertainty. We calculate the KL divergence of each journal’s Citation Distribution relative to the Overall Scientific Content Distribution, denoted as $\Delta_{0}$. This represents the uncertainty of a journal’s own research content relative to the knowledge environment. The KL divergence of each journal’s Citation Distribution relative to its Reference Distribution is denoted as $\Delta_{1}$. This represents the uncertainty of a journal’s own research content relative to its reference content. The KL divergence of each journal’s Reference Distribution relative to the Overall Scientific Content Distribution is denoted as $\Delta_{2}$. This represents the uncertainty of a journal’s reference content relative to the knowledge environment.(2)$$\begin{array}{cc} KL(P\|Q) & =\sum_{i}P\left(i\right)\log\left(\frac{P(i)}{Q(i)}\right)\\ \Delta_{0}\left(i\right) & =KL(d_{1}\left(i\right)\|d_{3})\\ \Delta_{1}\left(i\right) & =KL(d_{1}\left(i\right)\|d_{2}\left(i\right))\\ \Delta_{2}\left(i\right) & =KL(d_{2}\left(i\right)\|d_{3}) \end{array}$$

Physical systems tend toward a stable state of minimum energy. The origins of information theory are also closely related to thermodynamic entropy. This study views the citation selection behavior of journals as a process of spontaneously seeking the path of least “energy” consumption in an abstract “Cognitive Energy Landscape.” By constructing and perturbing this energy landscape, we can verify whether citation selection follows a “minimum energy” principle. For any given journal *i*, the “cognitive energy” *E* after selecting a set of references as intermediaries is effectively. As shown in Equation (Equation 3) this energy function measures the “total cognitive cost” required to connect its own content $d_{1}\left(i\right)$ with the knowledge environment $d_{3}$ by citing $d_{2}\left(i\right)$. And Mediating Similarity means that reference portfolio acts as a cognitive bridge to optimize the cognitive energy cost, shown in Figure 2.(3)$$E\left(i\right)=\Delta_{1}\left(i\right)+\Delta_{2}\left(i\right)$$

## 4. Results

To test whether the energy-minimizing effect of a reference portfolio is a generic property of any set of similar journals or a specific feature of the actual curated set, we conducted a simulation. We constructed hypothetical reference portfolios by selecting the *k* journals closest to a target journal, based on two different similarity metrics: KL divergence and Cosine similarity. We then plotted the total “cognitive energy” (the sum of $\Delta_{1}+\Delta_{2}$, aggregated over all 19,129 journals) as a function of the portfolio size, *k*, which was increased incrementally along a Fibonacci sequence.

Figure 3 displays the results when using KL divergence to measure the similarity between journals. The blue curve represents the cognitive energy of these synthetically constructed reference sets. It exhibits a distinct U-shape, illustrating a fundamental trade-off. When *k* is small, the reference portfolio is highly specialized and close to the target journal’s own topic (small $\Delta_{1}$), but it is a poor bridge to the broader scientific landscape (large $\Delta_{2}$). Conversely, when *k* is large, the portfolio becomes more diverse and closer to the global average (small $\Delta_{2}$), but it loses its specific relevance to the target journal (large $\Delta_{1}$). The horizontal dashed line at 80,939 units represents the total cognitive energy of the direct path ($\Delta_{0}$) for all journals. The fact that the blue curve dips below this line indicates that using a set of KL-divergence-closest journals as intermediaries is, indeed, more efficient than the direct path. Most importantly, the red dot marks the cognitive energy of the actual reference portfolios at the median size of real-world reference lists (*k* = 976). Its position, significantly below the blue curve, provides a key insight: the actual reference portfolio is not merely a collection of the most similar journals; it is a synergistically optimized set that achieves a much lower cognitive energy than a naively constructed portfolio of the same size based on simple KL-divergence proximity.

Figure 4 shows the same experiment but using Cosine similarity to select the *k* closest journals. While the curve also shows a U-shaped trend, it is positioned significantly higher than the curve in Figure 3. Notably, for most values of *k*, the cognitive energy of these portfolios is greater than the direct path energy ($\Delta_{0}$), indicating that an improperly chosen reference set can be counterproductive, increasing the cognitive distance rather than shortening it. This highlights that Cosine similarity is a less effective proxy for “cognitive relevance” in this context. The red dot, representing the actual reference portfolios, remains in the same position, now appearing even further below the curve. This starkly reinforces our finding: the curation process behind real-world reference selection achieves a level of optimization that far surpasses what can be achieved by simply aggregating journals based on a standard similarity metric like Cosine similarity.

Taken together, these two figures demonstrate that while the principle of mediation is valid, the specific composition of the reference portfolio is critical. The actual reference lists are not just good, they are highly optimized, outperforming even the best-performing synthetic portfolios constructed from the most relevant candidates identified by KL divergence.

The data reveals that a focal journal’s cited journals are highly concentrated among those with the closest KL divergence. As Table 1 shows, on average, 83.4% of a journal’s actual references are found within the top 3n closest journals (where *n* is the number of actual references), providing strong empirical justification for constructing our perturbation experiment’s candidate pool from this set. Importantly, the analysis confirmed that the larger candidate pool size does not materially affect the core conclusions, demonstrating the robustness of our approach. Therefore, for the perturbation experiments, this study used a candidate pool of size 3n, composed of the actual cited journals and twice that number of the closest un-cited journals.

This study selected twice the actual reference number closest un-cited journals and the actual cited journals to form a sample pool of three times the real citation number for perturbation experiments, how to construct candidate pool shown as Algorithm 1.
**Algorithm 1** Constructing the Candidate Pool for a Target Journal**Require:** Target journal $J_{\mathrm{target}}$; Set of all journals $J_{\mathrm{all}}$**Ensure:** Candidate pool $P_{\mathrm{candidate}}$** ** *   Step 1: Identify actual references and their count*  1:$R_{\mathrm{actual}}\leftarrow \mathrm{CitedBy}\left(J_{\mathrm{target}}\right)$  2:$n_{\mathrm{actual}}\leftarrow \left|R_{\mathrm{actual}}\right|$** ** *   Step 2: Identify all un-cited journals*  3:$J_{\mathrm{uncited}}\leftarrow J_{\mathrm{all}}\setminus \left(R_{\mathrm{actual}}\cup \left\{J_{\mathrm{target}}\right\}\right)$** ** *   Step 3: Calculate KL divergence to all un-cited journals*  4:D←∅  5:**for** each journal $J_{j}\in J_{\mathrm{uncited}}$ **do**   ▹$d_{1}\left(i\right)$ as citation distribution of journal *i* as per Equation (1)  6:      $\mathrm{dist}\leftarrow KL(d_{1}\left(J_{\mathrm{target}}\right)\|d_{1}\left(J_{j}\right))$  7:      $D\leftarrow D\cup \{\left(J_{j},\mathrm{dist}\right)\}$  8:**end for**** ** *   Step 4: Sort and select the closest un-cited journals*  9:$L_{\mathrm{sorted}}\leftarrow \mathrm{SortPairs}\left(D,\mathrm{key}=\mathrm{distance},\;\mathrm{order}=\mathrm{asc}\right)$  10:$k\leftarrow 2\times n_{\mathrm{actual}}$  11:$R_{\mathrm{closest}\_\mathrm{uncited}}\leftarrow \left\{J_{j}|\left(J_{j},{\mathrm{dist}}_{j}\right)\in L_{\mathrm{sorted}}\left[1\ldots k\right]\right\}$** ** *    Step 5: Combine to form the final candidate pool*  12:$P_{\mathrm{candidate}}\leftarrow R_{\mathrm{actual}}\cup R_{\mathrm{closest}\_\mathrm{uncited}}$  13:**return** $P_{\mathrm{candidate}}$

For each journal *i*, we calculated its “real cognitive energy” $E_{real}$ using its actual reference distribution $d_{2}\left(i\right)$. We then performed global and local “perturbations” on its reference list by randomly selecting journals from the candidate pool to test the constraints of the mediating similarity effect across different dimensions. The experiment consisted of two parts. The first part was a global perturbation, where a number of journals equal to the real citation count were randomly drawn from the sample pool to form an experimental sample. This was repeated 1000 times, and the energy consumed by these random combinations was compared with that of the real citation combination. The second part was a local perturbation, where the real citation combination was maintained, but 10% of its journals were replaced with journals drawn from the sample pool. This was also repeated 1000 times and compared with the energy of the real citation combination.

Our initial perturbation experiments, conducted on a sample of 500 top-ranked journals by the SJR indicator, revealed a strong optimization pattern. In the global perturbation test, the real cognitive energy ($E_{real}$) for 403 of these elite journals was lower than the energy of all 1000 randomly assembled portfolios. Figure 5 shows the distribution of percentile of global perturbation. Under local perturbation, $E_{real}$ was not always the lowest but was significantly lower than the mean of the locally perturbed energies, on average being lower than 77.25% of them. Figure 6 shows the distribution of percentile of local perturbation.

To formally address the need for greater statistical rigor and to test the universality of this principle beyond top-tier journals, we conducted a new, more extensive analysis on a random sample of 500 journals drawn from the entire dataset of 19,129. For this random sample, we calculated effect sizes (Cohen’s d) and bootstrapped 95% confidence intervals (CIs) from 10,000 resamples to support our findings. The results from this broader sample provide powerful statistical validation for the global optimization principle. The mean percentile rank of the real portfolio ($E_{real}$) within its 1000 randomly assembled alternatives was 1.34% (95% CI: [0.80%, 1.97%]). This confirms that actual reference portfolios are consistently found at the extreme low end of the cognitive energy distribution in global perturbation test. The effect size for this comparison was exceptionally large, with a mean Cohen’s d of 3.93 (95% CI: [3.79, 4.07]). The local perturbation experiment, which reflects a robust “satisficing” strategy, yielded similarly strong results on the random sample. The mean percentile rank of the real portfolio among its 1000 locally perturbed variations was 19.41% (95% CI: [18.59%, 20.24%]), demonstrating that under local perturbation test the actual portfolio is superior to the vast majority of slightly altered combinations. The corresponding effect size was a large Cohen’s d of 0.78 (95% CI: [0.76, 0.81]).

Collectively, the narrow confidence intervals from the random sample analysis confirm that our estimates are precise and the findings are robust and generalizable. The large effect sizes underscore the practical significance of the Mediating Similarity principle. To illustrate these dynamics for a specific, high-impact case, we present the global and local perturbation energy distributions for the journal Cell in Figure 7 and Figure 8, respectively.

The global resampling experiment shows that, on a macro scale, a journal’s real citation portfolio is a highly optimized whole, whose synergistic effect far exceeds the simple sum of relevant literature. However, the local replacement experiment further reveals that this optimization is not without constraints in the real world. Although the real citation portfolio shows strong local stability and robustness, a small number of combinations with lower energy may theoretically exist. This suggests that the final citation choice is a result of a compromise between the theoretical drive for “minimum energy” and real-world constraints like path dependence and discovery costs, reflecting a pursuit of a “satisficing” rather than an absolute “optimal” solution in scientific practice. We draw a diagram to visually represent this duality as Figure 9.

## 5. Discussion

This study introduced and validated the principle of “Mediating Similarity.” It demonstrates that a journal’s reference list acts as an optimized cognitive bridge between its unique scientific identity and the broader knowledge environment. A crucial question is how this principle relates to established sociological theories of citation behavior, namely the normative theory and the social constructivist view [21]. Our findings do not invalidate these classical perspectives. Instead, they suggest that “Mediating Similarity” operates at a more fundamental, almost “physical” level. It describes the underlying mechanics of information navigation, upon which social and normative behaviors are ultimately built.

The normative theory, rooted in the work of [22], frames citation as a moral obligation to acknowledge intellectual debt. It is a social norm that ensures credit is given where it is due, thereby maintaining the integrity of the scientific reward system. The social constructivist view, in contrast, sees citation as a rhetorical tool for persuasion [23]. From this perspective, authors cite strategically to bolster their arguments, align with powerful figures, and enhance the perceived authority of their work. Both theories focus on the conscious or subconscious motivations of the individual author, with one driven by duty and the other by self-interest and persuasion.

Our “Mediating Similarity” principle offers a complementary, yet distinct, perspective. It does not primarily concern itself with an author’s psychological or social motivations. Instead, it proposes a system-level organizing principle analogous to a physical law, like the principle of least action. We conceptualize the scientific knowledge landscape as a “Cognitive Energy Landscape.” In this landscape, “cognitive energy” represents the cost of connecting a specific research identity to the global knowledge environment, and it is measured by the sum of KL divergences. Our findings suggest that reference behavior is a process of spontaneously seeking the path of least “energy” consumption. A journal’s reference list, therefore, is not just a collection of acknowledgements or persuasive tools. It is the most efficient cognitive path discovered to bridge the informational distance between its own niche and the broader scientific universe.

A crucial clarification is warranted regarding our use of agent-based language. We do not claim that individual authors or editors are consciously calculating KL divergence to minimize “cognitive energy.” Rather, “Mediating Similarity” is a descriptive principle of an emergent, system-level phenomenon. This macroscopic pattern of energy minimization arises from the aggregation of countless microscopic, goal-oriented decisions. These decisions are made by individual researchers operating under bounded rationality. Authors select references based on practical heuristics, such as seeking relevance, building on prior work, justifying methods, and ensuring their arguments are comprehensible to peers. When an author cites papers to bridge their novel idea with established literature, they are, in information-theoretic terms, creating an efficient encoding of their position. This act of contextualization is repeated across thousands of papers. As a result, the collective reference portfolio of a journal becomes an efficient information-theoretic intermediary. Therefore, the “journal as an agent” in our model is not a literal decision-maker. It is an abstraction representing the aggregated citation practices of its community. Our framework is thus not an agent-based model in the traditional sense. Rather, it reveals a fundamental statistical regularity in the self-organization of scientific communication. For this phenomenon, the language of physics and agency provides a potent and predictive metaphor.

This “physical” interpretation has several important implications:

First, it provides a potential underlying mechanism for both normative and social behaviors. Why does acknowledging intellectual debt (normative view) feel “right”? Perhaps because citing foundational work correctly places one’s research within an established, low-energy cognitive trajectory. Why is citing authoritative papers (constructivist view) a good persuasive strategy? Because these papers often act as major hubs in the knowledge network. Citing them is an efficient way to connect to a large, relevant portion of the scientific landscape. In this sense, normative and social strategies may be successful precisely because they are effective heuristics for navigating the cognitive energy landscape we have described. Our principle does not replace these theories but provides a more fundamental, quantitative foundation for them.

Second, it highlights the importance of the reference portfolio as a holistic entity. Both normative and social constructivist theories tend to focus on the reasons for citing individual papers. Our global perturbation experiment, however, reveals that the value of a reference list lies in the synergistic effect of the entire combination, not just in the sum of its parts. The actual reference portfolio is a highly optimized whole. It is far more efficient at minimizing cognitive energy than any random assortment of individually relevant papers. This suggests that reference selection is a process of curating a “balanced diet” of knowledge, where the interplay between references is as important as the references themselves.

Third, it reconciles optimization with real-world constraints through the “satisficing” principle. Our local perturbation experiment showed that while the actual reference portfolio is highly robust, it is not always the absolute mathematical optimum. This aligns perfectly with Herbert Simon’s principle of satisficing [35]. In the complex and uncertain process of scientific discovery, authors do not perform an exhaustive search for the single “best” set of references. Instead, they find a “good enough” solution. This solution effectively minimizes cognitive distance within the practical constraints of path dependence, discovery costs, and serendipity. This reframes reference behavior not as a perfect optimization, but as a boundedly rational, yet highly effective, strategic navigation.

In summary, normative and social constructivist theories provide rich explanations for why authors cite. The “Mediating Similarity” principle offers a more fundamental, “physical” explanation for how the system of scientific knowledge organizes itself. It suggests that citation is not merely a social act but a strategic behavior, driven by the universal need to process information efficiently. By framing reference selection as the search for an optimal cognitive path in an energy landscape, our work advances the understanding of the self-organizing dynamics of science. It reveals a force-like mechanism that compels the scientific community to build the most efficient bridges across the vast terrain of human knowledge.

## 6. Conclusions

This study reframes the fundamental nature of reference behavior, moving beyond sociological motivations to uncover an information-theoretic principle of self-organization within science. We have empirically established the “Mediating Similarity” principle, demonstrating that a journal’s reference portfolio is not a passive list of influences but a synergistically optimized construct. It functions as the most efficient cognitive bridge connecting a specialized research area to the vast landscape of global knowledge. Our findings reveal that this efficiency arises from a powerful dual mechanism: a drive toward global optimization, where the actual reference set as a whole vastly outperforms randomly assembled but relevant alternatives, tempered by the realities of local satisficing, where the selection process operates within the practical constraints of discovery and path dependence. This duality resolves a long-standing tension: how the scientific enterprise achieves such remarkable coherence and progress while being composed of countless individual, boundedly rational decisions. The answer, our work suggests, lies in a system-level tendency to minimize “cognitive energy.” The reference portfolio as a holistic entity—not just the sum of its parts—is the key unit of selection in this process. Beyond its theoretical contribution, the “Mediating Similarity” principle offers tangible implications for the practice and evaluation of science. It provides a new dimension for assessing a journal’s role and quality. Beyond traditional impact metrics that measure attention received, we can now quantify a journal’s “integrative capacity”—its efficiency in bridging its niche with the broader scientific environment. This could identify journals that are crucial hubs of knowledge integration, even if they are not the most highly cited. This principle challenges the current paradigm of recommending individual papers. Future tools could be designed to recommend a balanced portfolio of references, helping authors construct a more robust cognitive bridge and better position their work for impact by suggesting a synergistic mix of foundational, contextual, and novel citations. Editors, reviewers, and funding agencies can use this framework as a conceptual tool. It provides a structured way to assess whether a manuscript or proposal is too insular or, conversely, effectively integrated into the relevant scientific discourse, thereby fostering more robust and potentially transformative interdisciplinary research. In conclusion, this work provides a foundational, “physical” law that governs the architecture of scientific communication. By framing reference selection as a strategic process of navigating a cognitive energy landscape, we open new avenues for research into the temporal evolution of scientific fields, the emergence of disciplines, and the dynamics of innovation. Science does not just accumulate knowledge; it actively and efficiently weaves it together. Our findings reveal a fundamental force driving that process, compelling the scientific community to build the most efficient bridges across the vast terrain of human knowledge.

## Figures and Tables

**Figure 1 entropy-27-01124-f001:**
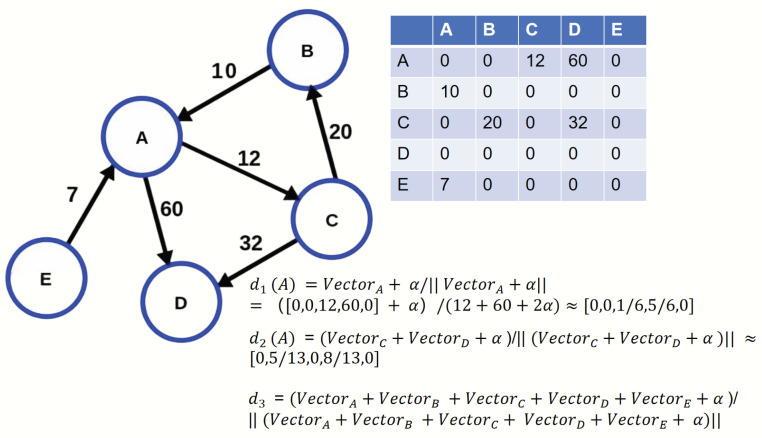
A toy model demonstrating three distributions.

**Figure 2 entropy-27-01124-f002:**
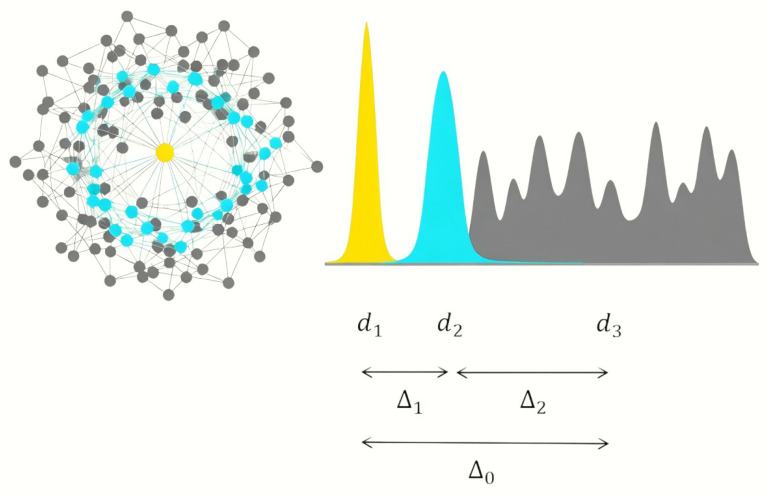
Illustration of the Mediating Similarity principle.

**Figure 3 entropy-27-01124-f003:**
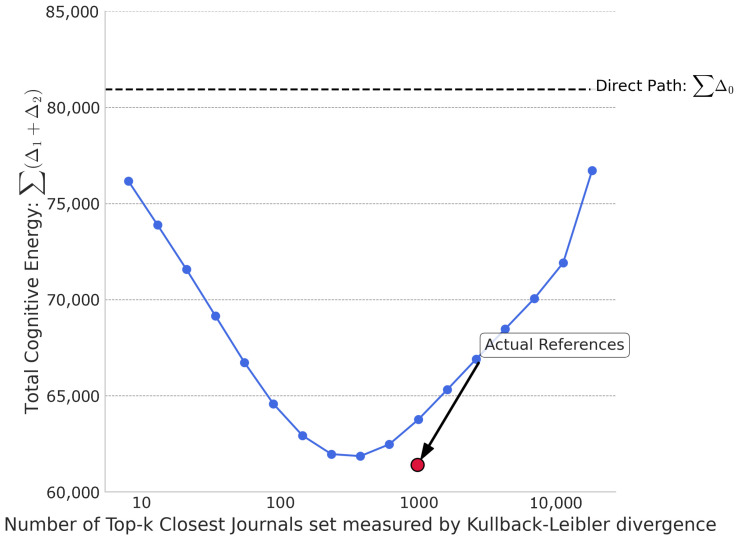
Cognitive energy landscape vs. closest journals reference set size measured by KL divergence.

**Figure 4 entropy-27-01124-f004:**
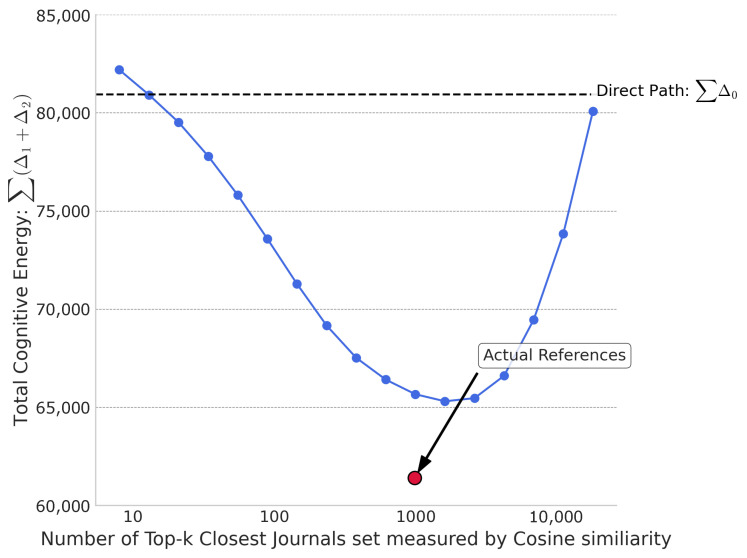
Cognitive energy landscape vs. closest journals reference set size measured by cosine similarity.

**Figure 5 entropy-27-01124-f005:**
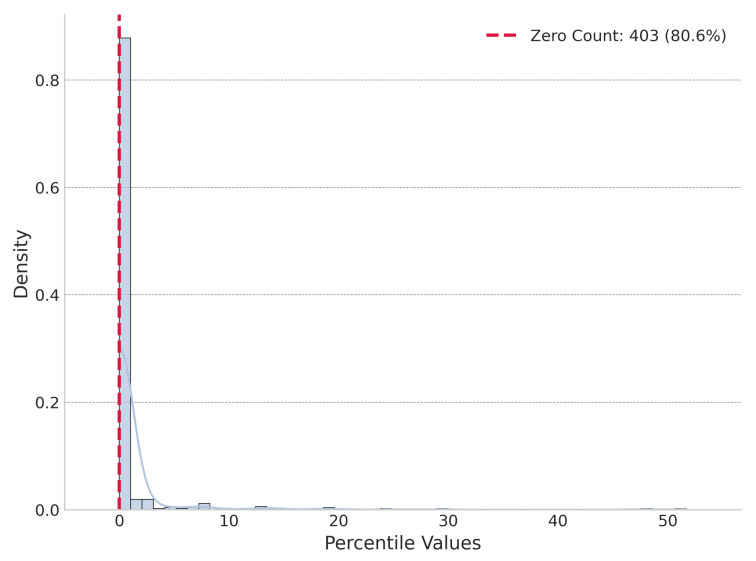
Distribution of percentile for global perturbation of top-ranked journals.

**Figure 6 entropy-27-01124-f006:**
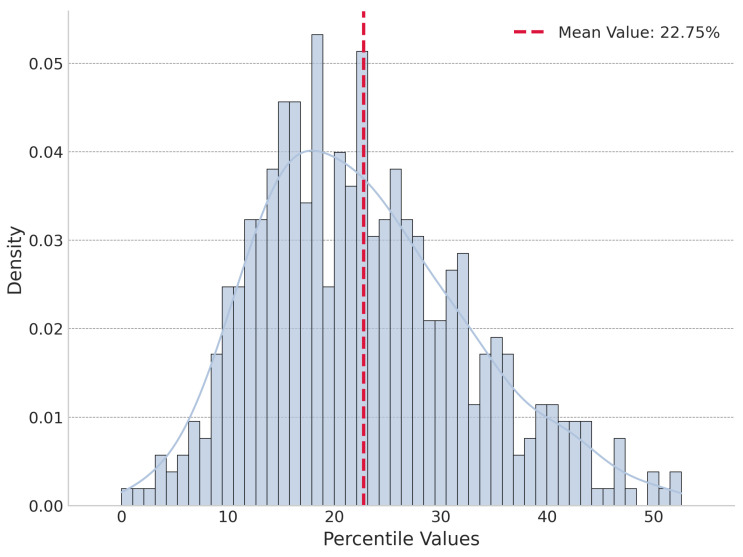
Distribution of percentile for local perturbation of top-ranked journals.

**Figure 7 entropy-27-01124-f007:**
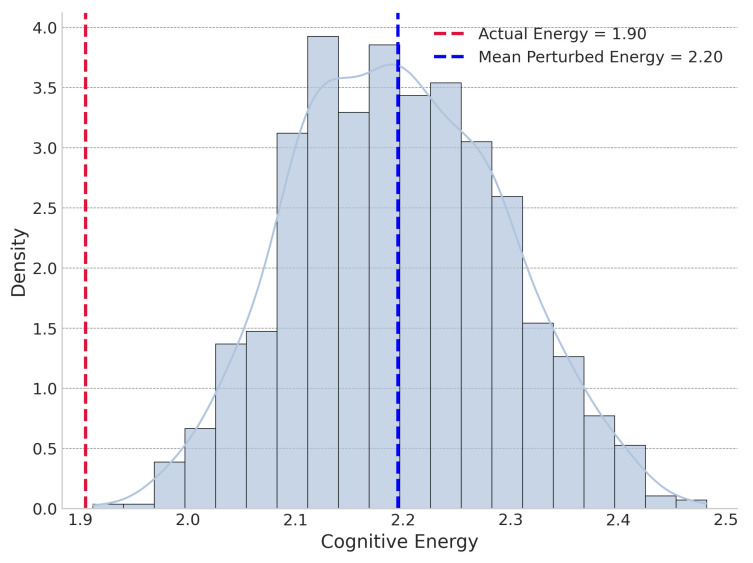
Global perturbation cognitive energy distribution for Cell.

**Figure 8 entropy-27-01124-f008:**
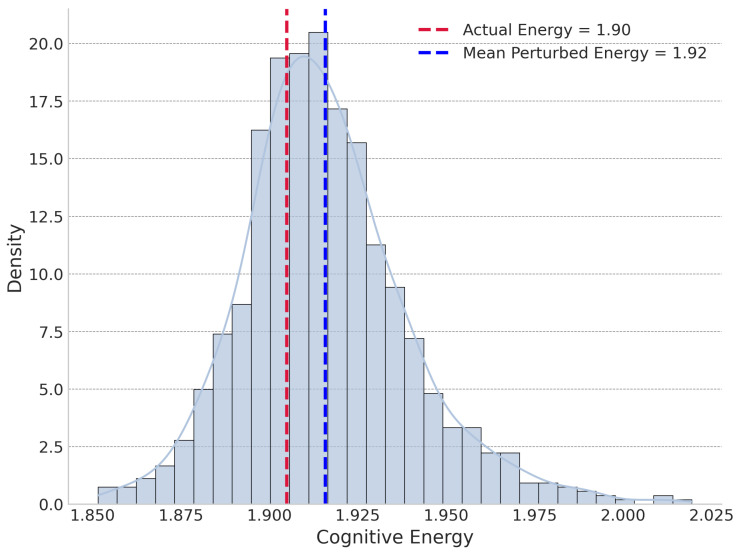
Local perturbation cognitive energy distribution for Cell.

**Figure 9 entropy-27-01124-f009:**
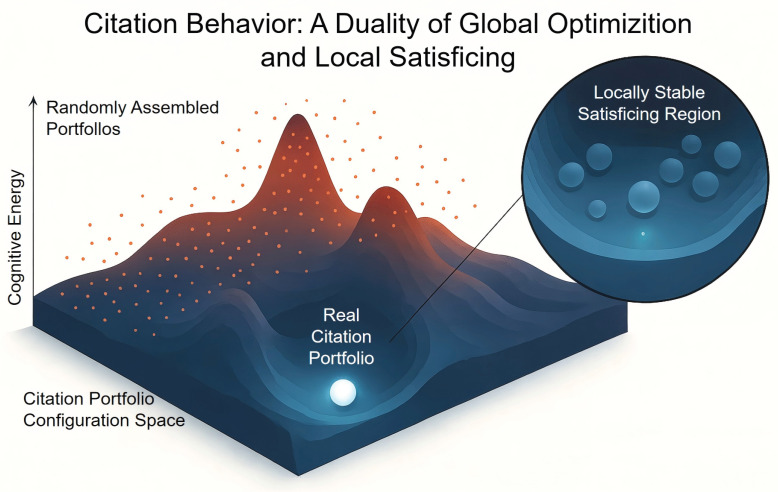
Duality of Global Optimization and Local Satisficing.

**Table 1 entropy-27-01124-t001:** The percent of closest journals measured by KL divergence or Cosine similarity in real reference journals.

	Within 1× the Actual Reference Number	Within 2×	Within 3×
KL divergence	0.588	0.764	0.834
Cosine similarity	0.430	0.618	0.727

## Data Availability

Data available at: https://doi.org/10.5281/zenodo.16903123, and code can be found at https://github.com/1748524327z/Mediating-Similarity (accessed on 24 October 2025).

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
