# Peer review of "Mediating Similarity: An Information-Theoretic Principle of Reference Behavior"

_entropy, 2025, doi:10.3390/e27111124_

Round 1

Reviewer 1 Report

Comments and Suggestions for Authors

This manuscript proposes a novel and compelling theoretical framework—mediated similarity—and provides empirical support using large-scale data and perturbation experiments. It possesses high academic value and potential impact, particularly in advancing understanding of citation behavior in scientometrics. The authors hope to further improve the manuscript by implementing the following revisions.
1. The manuscript does not clearly explain how to handle zero grids; the authors are requested to provide further details in the Methods section.
2. The authors should provide detailed information on the candidate library selection rules and parameters.
3. If possible, to prevent conclusions from being biased by a few major journals or disciplines, the authors may consider conducting subsample analyses by major discipline or SJR, and stratifying by citation count.
4. The authors are advised to exclude or annotate "journal self-citations." If this is not possible within a short period of time, the authors are advised to explain their potential impact on the conclusions in the limitations section.

Author Response

This manuscript proposes a novel and compelling theoretical framework—mediated similarity—and provides empirical support using large-scale data and perturbation experiments. It possesses high academic value and potential impact, particularly in advancing understanding of citation behavior in scientometrics. The authors hope to further improve the manuscript by implementing the following revisions.

  1. The manuscript does not clearly explain how to handle zero grids; the authors are requested to provide further details in the Methods section.

Respond:

We are grateful to the reviewer for pointing out this important omission. The reviewer is correct that our manuscript did not sufficiently detail how we handled zero-value entries in our vectors, which is a critical step for calculating Kullback-Leibler (KL) divergence.

To address this, we employed an additive smoothing technique. Specifically, before normalizing the raw citation count vectors, we added a small constant (α = 1e-9) to every count. This ensures that all probabilities are non-zero, thus guaranteeing the numerical stability of the KL divergence calculations while having a negligible impact on the distribution's structure.

We have now revised the Methods section to explicitly describe this process. We believe this addition improves the clarity and reproducibility of our work.

  1. The authors should provide detailed information on the candidate library selection rules and parameters.

Response:

We thank the reviewer for this important question, which allows us to clarify the selection criteria for the journals in our study.

First, regarding the overall set of journals for our analysis, we selected them from the 2021 SCImago Journal Rank (SJR) list. As we explain in Section 3.1 (Data), we chose the SJR list for two main reasons: 1) its public accessibility ensures the transparency and reproducibility of our selection process, and 2) its high correlation with other major journal metrics ensures that our sample is representative and robust, without introducing significant selection bias. Of the 27,339 journals on the 2021 SJR list, we identified 19,129 in the OpenAlex database, and this set formed the basis for our network construction and experiments.

Second, for the perturbation experiments, we constructed a specific "candidate pool" for each target journal. This pool was designed to include highly relevant potential references. The rule was to create a pool containing the journal's actual references plus the top closest un-cited journals, where closeness was measured by KL divergence. The size of this pool was set to three times the number of the journal's actual references. The detailed algorithm for this process is provided in Algorithm 1 on page 10.

We hope this clarifies the selection rules and parameters for both the overall study and the specific experiments.

  1. If possible, to prevent conclusions from being biased by a few major journals or disciplines, the authors may consider conducting subsample analyses by major discipline or SJR, and stratifying by citation count.

Response:

We are very grateful to the reviewer for this constructive and insightful suggestion. The concern that our findings might be biased by a small number of high-impact journals or specific disciplines is a critical point, and we appreciate the opportunity to address it more rigorously.

Our initial analysis focused on a sample of 500 top-ranked journals based on the SJR indicator. The primary reason for this choice was the high computational complexity of the perturbation experiments. Each experiment for a single journal involves thousands of iterations and KL divergence calculations on high-dimensional vectors, making it infeasible to run on the entire population of 19,129 journals. We initially reasoned that top-tier journals, as exemplars of scientific communication, would serve as strong representatives for demonstrating the principle of "Mediating Similarity."

However, we fully agree with the reviewer that this approach could potentially introduce bias and that verifying the universality of our principle is essential. Following the reviewer's excellent advice, we have conducted a new, more extensive analysis on a random sample of 500 journals drawn from the entire dataset of 19,129. This new experiment is designed specifically to mitigate potential biases related to journal rank, discipline, or citation count, thereby testing the generalizability of our findings.

The results from this new analysis on the random sample provide powerful statistical validation for our principle and confirm that the observed effects are not confined to elite journals. We have added a detailed report of these findings to the Results section (page 11, lines 345-360). The key results are:

Global Perturbation: For the random sample, the mean percentile rank of a real reference portfolio was 1.34% (95% CI: [0.80%, 1.97%]) compared to its randomly assembled alternatives. The effect size was exceptionally large (mean Cohen’s d = 3.93, 95% CI: [3.79, 4.07]).

Local Perturbation: The mean percentile rank was 19.41% (95% CI: [18.59%, 20.24%]), with a large effect size (mean Cohen’s d = 0.78, 95% CI: [0.76, 0.81]).

These findings from the random sample are consistent with our initial results but provide much stronger evidence for the universality of the "Mediating Similarity" principle across the entire scientific landscape. The narrow confidence intervals and large effect sizes underscore the robustness and practical significance of our conclusions.

We believe that this new experiment, prompted by the reviewer's valuable suggestion, substantially strengthens our manuscript by demonstrating that our conclusions are not an artifact of sampling bias. We thank the reviewer again for pushing us to improve the rigor of our work.

  1. The authors are advised to exclude or annotate "journal self-citations." If this is not possible within a short period of time, the authors are advised to explain their potential impact on the conclusions in the limitations section.

Response:

We thank the reviewer for raising this very important point regarding journal self-citations. This is a crucial methodological consideration, and we appreciate the opportunity to clarify our approach and explain why self-citations do not introduce a significant bias that would alter our conclusions.

While we did include journal self-citations in our analysis, we argue that their effect on our findings is structurally negligible due to the specific design of our information-theoretic measures. We will elaborate on two key points: the dilution effect inherent in our methodology and the conceptual validity of including self-citations.

  1. The Methodological Dilution Effect:

The core of our argument lies in how the Journal's Reference Distribution (d2) is constructed. As defined in Section 3.2 and Equation 1, d2 is not a simple average of its constituent reference profiles; it is the normalized sum of the citation vectors of all journals cited by the target journal.

Let's consider a practical example. The median number of unique journals referenced by a single journal in our dataset is 976. When a journal cites itself, its own citation vector (V_self) becomes just one of these 976 vectors that are summed to create the aggregate vector for d2. The final d2 distribution is then calculated as:

d2 = normalize(V_ref1 + V_ref2 + ... + V_self + ... + V_ref976)

Therefore, the contribution of the single self-citation vector is heavily diluted by the contributions of the hundreds of other cited journals. The resulting d2 distribution is overwhelmingly shaped by the collective profile of the entire reference portfolio, not by the single self-citation. The low value of KL(d1 || d2) is not an artifact of the self-citation making d2 trivially similar to d1. Instead, it reflects the fact that the entire portfolio of hundreds of cited journals collectively forms a distribution that is a good proxy for the citing journal's own position. The self-citation's contribution to this is minimal.

  1. The Conceptual Validity of Including Self-Citations:

From a conceptual standpoint, journal self-citation is a natural and integral part of scientific communication. Journals often build upon their own prior work, and these citations reflect legitimate intellectual continuity and specialization within a field. Excluding them would artificially remove this valid signal and present an incomplete picture of a journal's reference behavior. Our framework aims to model the real-world process of reference selection, and self-citation is a genuine part of that process.

Given the strong methodological dilution effect and the conceptual validity of including this behavior, we are confident that journal self-citations do not introduce a bias that would compromise our conclusions about the "Mediating Similarity" principle. The synergistic optimization we observe is a feature of the entire portfolio as a holistic entity, an effect far too strong to be driven by a single, diluted data point.

Therefore, we believe that neither excluding self-citations (which would be methodologically complex and conceptually questionable) nor adding a specific note in the limitations section is necessary, as the effect is structurally controlled by the method itself. We hope this explanation adequately addresses the reviewer's valid concern.

Reviewer 2 Report

Comments and Suggestions for Authors

Dear Authors,

Thank you for the opportunity to review your manuscript titled “Mediating Similarity: An Information-Theoretic Principle of Reference Behavior.” This is an ambitious and intriguing paper that applies information theory to scientometric analysis in a novel way. The central claim — that reference lists serve as optimized cognitive bridges between a journal’s citation identity and the broader knowledge system — is compelling and timely. However, while the manuscript presents interesting results, there are several areas where clarity, structure, and methodological precision could be improved to strengthen the impact and reproducibility of the study.

Below are section-wise comments for your consideration:

Abstract

  • Sentence Clarity & Overloading: The abstract is dense and includes too many technical details (e.g., exact journal counts, perturbation test descriptions, specific KL divergence metrics). These can overwhelm the reader and dilute the main message. Consider condensing and simplifying while emphasizing the core insight. Focus on clearly stating the research question, the proposed principle (“Mediating Similarity”), the methods used, and key findings. Move details like exact journal counts or experimental ratios to the main text.

Introduction

  • Line 99–103 ("Mathematically, this means..."): The KL divergence inequality is critical to the paper’s thesis. However, this is introduced rather abruptly. Consider offering a brief intuitive explanation before presenting the formal inequality.
  • Repetition: Much of the theoretical foundation is repeated later in the "Related Works" section. Consider consolidating overlapping ideas to reduce redundancy.
  • Minor Terminology Clarification: At times, the term "Citation Distribution" is used interchangeably with "identity" or “scientific content” without early explanation. A clearer early definition of all three distributions (citation, reference, and overall) would help.

Related Works

  • Too Broad and Unfocused: While this section is rich in references, it occasionally strays too far into generalized commentary on entropy and interdisciplinarity (e.g., lines 144–173). This weakens its function as a focused literature review tied to your specific research problem.
  • Missing Gaps in Literature: While foundational studies are cited, there's limited discussion of existing work that directly explores the mechanics of reference selection or strategic citation behavior. Including recent studies on citation modeling or behavioral bibliometrics would make your contribution stand out more clearly. Refocus this section to more explicitly highlight the research gap your study addresses.

Methods

  • Lack of Justification for Key Choices:
    • Why was the SCImago Journal Rank (SJR) used over other bibliometric rankings?
    • Why were the top 500 journals chosen for perturbation experiments? Was this threshold empirically or conceptually motivated?
  • Equation Presentation (Eq. 1–3):
    • The equations are central but presented with inconsistent formatting and minimal explanation. Variables like Vi, ni, and others need to be defined more carefully.
    • A diagram or schematic of the three distributions (d1, d2, d3) would be helpful to accompany the equations.
  • Ambiguity in Vector Operations: Expressions like Vi/||Vi|| assume familiarity with vector calculus, which may not be universal among readers of Entropy. Consider briefly clarifying the vector normalization process in plain terms.

Results

  • Over-Reliance on Figures Without Sufficient Interpretation
    Figures 2 and 3 are central to the paper’s argument, but their interpretation is underdeveloped. For example, the meaning of the red dot and the implications of its position relative to the fitted curves are not clearly explained. Likewise, the global and local perturbation experiments are presented in dense technical detail without a clear summary or illustrative example to guide the reader. Expand figure captions and in-text discussion to clearly explain the visual elements and their significance.
  • Statistical Rigor is Insufficient
    The claim that “99% of 1,000 randomly assembled portfolios” have higher cognitive energy than the real ones is compelling, but no formal statistical tests are reported. There are also no confidence intervals or effect sizes provided. Suggestion: Include formal hypothesis testing (e.g., Wilcoxon signed-rank, permutation tests), report p-values and effect sizes, and provide bootstrapped confidence intervals to support key results.
  • Reproducibility Gaps
    Key details about how “closeness” in KL divergence was calculated are missing. It is unclear how journals were selected for candidate pools in the perturbation experiments and whether any thresholds or filters were applied (e.g., to exclude low-citation journals). The number of iterations (1,000) is also not justified. Suggestion: Clarify the methods for ranking journal similarity, defining candidate pools, and conducting perturbations. Provide algorithmic or pseudocode-level descriptions if possible.
  • Limited Generalizability
    The perturbation tests focus solely on the top 500 journals by SJR, which could bias the results toward elite or multidisciplinary journals. Suggestion: Extend the analysis to a broader or stratified sample of journals across different disciplines and impact levels to test the robustness of the findings.
  • Terminology Overload
    Phrases like “cognitive energy landscape,” “global resampling,” and “satisficing optimization” are used without clear definition. While metaphorically appealing, they require clarification to avoid confusion. Suggestion: Define key terms clearly when first introduced and distinguish between technical concepts and metaphorical framing.

Discussion

  • Novelty Needs Tighter Framing:
    • While the notion of "Mediating Similarity" is original, the discussion could benefit from tighter comparisons to alternative hypotheses (e.g., random citation behavior, reputation bias).
    • The implication that journals behave like agents minimizing cognitive energy is metaphorically rich, but the paper does not clarify whether this is a descriptive pattern or an agent-based modeling claim.
  • Caution Needed:
    • Some statements verge on overclaiming, e.g., line 344–345 “...this phenomenon is not an artifact but a fundamental and universal feature.” You could qualify these claims more carefully — e.g., within the scope of the dataset used.

Conclusion

  • Restates Results Rather Than Synthesizes: The conclusion reiterates the experiments but does not clearly articulate how this changes our understanding of scientometrics or citation behavior.
  • Missed Opportunity for Broader Implications: Consider discussing potential applications — e.g., how this understanding could be used in journal evaluation, recommender systems, or editorial policy.

Figures and Tables

  • Figure 1 (Cognitive Bridge Illustration): The figure is not labeled in detail nor referenced clearly in the results. Consider improving its caption and explicitly walking the reader through it.
  • Figures 2–3: Ensure axes are clearly labeled (e.g., ∆1, ∆2 units), and that all visual elements (like the red dot) are described fully in the legend or text.
  • Table 1: While informative, this table would be stronger if compared with a random baseline or control.

Writing and Language

  • Typographical Errors:
    • Line 100: “ciatation” → “citation”
    • Line 24: “averagely” is unidiomatic in English — consider “on average”
  • Redundancy: Several ideas are repeated almost verbatim across sections (e.g., lines 95–98 and 205–208). A tighter structure would enhance flow.
  • Tone: Generally clear and academic, though slightly more engagement with potential critiques (e.g., limits of KL divergence as a proxy for cognition) would be valuable.

Data and Code Availability

  • The Zenodo dataset is cited (line 390), but there is no reference to accompanying code or procedures that would enable reproduction of the perturbation experiments. If possible, release the code or explain the steps in a supplementary appendix.

Author Response

Dear Authors,

Thank you for the opportunity to review your manuscript titled “Mediating Similarity: An Information-Theoretic Principle of Reference Behavior.” This is an ambitious and intriguing paper that applies information theory to scientometric analysis in a novel way. The central claim — that reference lists serve as optimized cognitive bridges between a journal’s citation identity and the broader knowledge system — is compelling and timely. However, while the manuscript presents interesting results, there are several areas where clarity, structure, and methodological precision could be improved to strengthen the impact and reproducibility of the study.

Below are section-wise comments for your consideration:

Abstract

Sentence Clarity & Overloading: The abstract is dense and includes too many technical details (e.g., exact journal counts, perturbation test descriptions, specific KL divergence metrics). These can overwhelm the reader and dilute the main message. Consider condensing and simplifying while emphasizing the core insight. Focus on clearly stating the research question, the proposed principle (“Mediating Similarity”), the methods used, and key findings. Move details like exact journal counts or experimental ratios to the main text.

Response:

We thank the reviewer for this valuable suggestion. We agree that the original abstract was overly dense with technical specifics.

Following your advice, we have rewritten the abstract to be more focused and accessible. The revised version omits the detailed experimental parameters and instead highlights our central research question, the "Mediating Similarity" principle, and our key findings. We believe the abstract now more effectively communicates the core contribution of our study.

Introduction

Line 99–103 ("Mathematically, this means..."): The KL divergence inequality is critical to the paper’s thesis. However, this is introduced rather abruptly. Consider offering a brief intuitive explanation before presenting the formal inequality.

Response:

We thank the reviewer for this valuable feedback. We agree that a more intuitive explanation was needed before presenting the formal inequality.

We have revised this section to first explain the concept in terms of informational "distance," clarifying that the mediated path through references is shorter than the direct path. The KL divergence inequality is then introduced as the formal expression of this core idea. We believe this improves the clarity and flow of the argument.

Repetition: Much of the theoretical foundation is repeated later in the "Related Works" section. Consider consolidating overlapping ideas to reduce redundancy.

Response:

Thank you for this suggestion. We have revised the Introduction and Related Works sections to avoid redundant overlap and improve the flow of the manuscript.

Minor Terminology Clarification: At times, the term "Citation Distribution" is used interchangeably with "identity" or “scientific content” without early explanation. A clearer early definition of all three distributions (citation, reference, and overall) would help.

Response:

Thank you for this important point. We agree that the interchangeable use of these terms could cause confusion.

We have revised the manuscript to provide clear and explicit definitions for "Journal's Citation Distribution," "Journal's Reference Distribution," and "Overall Scientific Content Distribution" early in the text (Introduction, lines 88-92). We now also clarify that the "Citation Distribution" serves as our formal proxy for a journal's "scientific identity." We believe this resolves the ambiguity.

Related Works

Too Broad and Unfocused: While this section is rich in references, it occasionally strays too far into generalized commentary on entropy and interdisciplinarity (e.g., lines 144–173). This weakens its function as a focused literature review tied to your specific research problem.

Missing Gaps in Literature: While foundational studies are cited, there's limited discussion of existing work that directly explores the mechanics of reference selection or strategic citation behavior. Including recent studies on citation modeling or behavioral bibliometrics would make your contribution stand out more clearly. Refocus this section to more explicitly highlight the research gap your study addresses.

Response:

We thank the reviewer for this insightful and constructive critique of the "Related Works" section. We agree that the original version was not sufficiently focused and did not adequately highlight the specific research gap our work addresses.

In response, we have undertaken a significant revision of this section. Following your advice:

We have streamlined the discussion, removing some of the broader, less relevant literature on general entropy applications.

We have incorporated a more targeted review of the literature on citation motivations, specifically discussing the normative and social constructivist theories, to better contextualize our work within behavioral bibliometrics.

This restructuring allows us to more explicitly frame the research gap: while existing work provides rich taxonomies for why individual citations are made, it lacks a unifying, system-level principle that explains the emergent logic governing the selection of a reference portfolio as a whole.

We are confident that the revised section now provides a much clearer and more compelling justification for our study and better highlights its unique contribution.

Methods

Lack of Justification for Key Choices:

Why was the SCImago Journal Rank (SJR) used over other bibliometric rankings?

Response:

We thank the reviewer for this question. As per your suggestion, we have clarified our rationale for using the SCImago Journal Rank (SJR) in the Data section (lines 205-211).

Our choice was based on two primary factors:

Transparency and Reproducibility: The SJR list is publicly accessible, which ensures our journal selection process is transparent and can be easily reproduced by other researchers.

Representativeness and Lack of Bias: As we note in the text, numerous major journal evaluation metrics (such as Journal Impact Factor, CiteScore, and SJR) are highly correlated. Therefore, using the widely recognized SJR list provides a representative and robust sample of reputable journals without introducing significant selection bias compared to using other established indicators.

We believe this justification confirms that our choice of SJR is methodologically sound for the purposes of this study.

Why were the top 500 journals chosen for perturbation experiments? Was this threshold empirically or conceptually motivated?

Response:

We thank the reviewer for this important question regarding our experimental design. Our initial choice of the top 500 SJR-ranked journals was motivated by a combination of conceptual and practical considerations.

Conceptual Motivation: We hypothesized that if the "Mediating Similarity" principle represents an optimization strategy, its effects would be most pronounced and detectable in the most established and high-impact journals, which serve as exemplars in the scientific ecosystem. Our initial experiment was therefore designed as a strong test of the principle in a cohort where we expected the signal to be clearest.

Practical Constraints: The perturbation experiments are computationally intensive. As described in the Methods section, for each journal, we performed 1,000 global and 1,000 local perturbations. The full set of experiments for this initial sample of 500 journals required nearly a full week of computation time. The number 500 was therefore chosen as a balance between obtaining a substantial sample for this initial test and maintaining a feasible runtime.

However, we fully agree with the reviewer's implicit concern that focusing solely on top-tier journals could limit the generalizability of our findings. To address this potential selection bias and to test the universality of the principle, we conducted a new, more rigorous analysis.

As now detailed in the Results section (lines 345-360), we performed the same perturbation experiments on a random sample of 500 journals drawn from the entire dataset of 19,129. The results from this random sample strongly validated our initial findings and confirmed the principle's universality. For instance, the mean percentile rank of the real portfolio in the global perturbation test was 1.34% (95% CI: [0.80%, 1.97%]), with a very large effect size (Cohen’s d = 3.93).

We believe this two-stage approach—an initial exploratory analysis on top journals followed by a robust validation on a random sample—provides a much stronger and more convincing case for the generalizability of our conclusions.

Equation Presentation (Eq. 1–3):

The equations are central but presented with inconsistent formatting and minimal explanation. Variables like Vi, ni, and others need to be defined more carefully.

A diagram or schematic of the three distributions (d1, d2, d3) would be helpful to accompany the equations.

Ambiguity in Vector Operations: Expressions like Vi/||Vi|| assume familiarity with vector calculus, which may not be universal among readers of Entropy. Consider briefly clarifying the vector normalization process in plain terms.

Response:

Thank the reviewer for these detailed and constructive comments regarding our mathematical presentation. We agree that the original formulation was not as clear or accessible as it should be.

In response, we have undertaken a thorough revision of the Methods section (lines 214-279) to address all the points raised:

Simplification and Clarification of Equations: To resolve the ambiguity of the vector operations and make the method more transparent, we have replaced the abstract vector norm notation (e.g., ||Vi||) with explicit summation forms (e.g., ∑k vi,k). This removes the reliance on vector calculus and presents the normalization process in a more direct, arithmetic manner.

Improved Variable Definitions and Formatting: We have carefully revised the notation and subscripts for all variables to ensure they are consistent, clearly defined in the text preceding the equations, and easy for the reader to follow.

Integration of Visual Aid: We have ensured that Figure 1, which provides a toy model, is now better integrated with the text to serve as the requested schematic. The text now more clearly directs the reader to this figure as a concrete example of how the three distributions (d1, d2, and d3) are calculated from the network structure.

Collectively, these changes make the mathematical formulation of our method significantly more transparent and self-contained. We believe the revised presentation now effectively communicates our methodology to the broad readership of Entropy.

Results

Over-Reliance on Figures Without Sufficient Interpretation

Figures 2 and 3 are central to the paper’s argument, but their interpretation is underdeveloped. For example, the meaning of the red dot and the implications of its position relative to the fitted curves are not clearly explained. Likewise, the global and local perturbation experiments are presented in dense technical detail without a clear summary or illustrative example to guide the reader. Expand figure captions and in-text discussion to clearly explain the visual elements and their significance.

Response:

We thank the reviewer for this critical feedback. We agree completely that the interpretation of our core results, particularly Figures 3 and 4 and the subsequent perturbation experiments, required more detailed explanation to be fully accessible and impactful.

In response, we have substantially revised the Results section (lines 280-319) to provide the detailed exposition you recommended.

Specifically for Figures 3 and 4, we have expanded the in-text discussion to clearly explain:

The meaning of the U-shaped curve: We now explicitly describe the fundamental trade-off it represents—how a small reference portfolio (k) is highly specialized but a poor bridge to the broader landscape, while a large portfolio is a better bridge but loses specific relevance.

The significance of the red dot: We have clarified that this dot marks the cognitive energy of the actual, real-world reference portfolios at the median reference list size.

The implication of its position: We now emphasize that its position significantly below the blue curve is the key insight. This visually demonstrates that actual reference lists are not merely collections of the most similar journals (as represented by the curve) but are synergistically optimized combinations that achieve a much lower cognitive energy.

Statistical Rigor is Insufficient

The claim that “99% of 1,000 randomly assembled portfolios” have higher cognitive energy than the real ones is compelling, but no formal statistical tests are reported. There are also no confidence intervals or effect sizes provided. Suggestion: Include formal hypothesis testing (e.g., Wilcoxon signed-rank, permutation tests), report p-values and effect sizes, and provide bootstrapped confidence intervals to support key results.

Response:

We thank the reviewer for this crucial point. We completely agree that our initial presentation of the perturbation results, while suggestive, lacked the formal statistical rigor required to make our claims robust. Your suggestion to incorporate formal statistical measures was extremely helpful.

In response, we have conducted a new, more extensive analysis and have updated the Results section (lines 345-360) accordingly. As you recommended, we have moved beyond simple percentile reporting to provide a more complete statistical picture:

Effect Size Estimation: To quantify the magnitude of the optimization effect, we calculated Cohen's d for the difference between the real portfolio's energy and the distribution of perturbed energies. The results show an exceptionally large effect for the global perturbation (mean Cohen's d = 3.93) and a large effect for the local perturbation (mean Cohen's d = 0.78).

Bootstrapped Confidence Intervals: To ensure the reliability and precision of our estimates, we generated 95% confidence intervals (CIs) for both the mean percentile ranks and the effect sizes, using 10,000 bootstrap resamples. For example, the mean percentile rank of the real portfolio in the global test is 1.34% with a very narrow 95% CI of [0.80%, 1.97%].

The percentile rank itself functions as a direct result from a permutation test, where a rank of 1.34% is statistically significant. By adding effect sizes and confidence intervals, we now provide a much richer assessment of the results.

These additions provide powerful statistical validation for our findings. The large effect sizes underscore the practical significance of the "Mediating Similarity" principle, while the narrow confidence intervals confirm that our findings are precise, robust, and generalizable. We are confident that this new, more rigorous statistical treatment substantially strengthens our conclusions.

Reproducibility Gaps

Key details about how “closeness” in KL divergence was calculated are missing. It is unclear how journals were selected for candidate pools in the perturbation experiments and whether any thresholds or filters were applied (e.g., to exclude low-citation journals). The number of iterations (1,000) is also not justified. Suggestion: Clarify the methods for ranking journal similarity, defining candidate pools, and conducting perturbations. Provide algorithmic or pseudocode-level descriptions if possible.

Response:

We thank the reviewer for this excellent suggestion. We agree that the methodological details for our perturbation experiments were not sufficiently clear, and your recommendation to add pseudocode was invaluable for improving the paper's transparency and reproducibility.

In response, we have added Algorithm 1: Constructing the Candidate Pool for a Target Journal (lines 324-326 and the subsequent algorithm block). This algorithm, along with the accompanying text, now explicitly addresses the points you raised:

Calculating "Closeness" and Ranking Similarity: Step 3 of the algorithm (Calculate KL divergence to all un-cited journals) now makes the process explicit. It shows that for a given target journal, we calculate the KL divergence from its citation distribution (d1(Jtarget)) to the distribution of every other un-cited journal. Step 4 then clarifies that these journals are sorted by this divergence value to create a ranked list of "closeness."

Defining the Candidate Pool: The full algorithm provides a step-by-step definition. It shows that the candidate pool is constructed by taking the union of two sets: (1) the set of the journal's actual references (Ractual), and (2) the set of the 2n closest un-cited journals, where n is the number of actual references. This results in a well-defined candidate pool of size 3n, as explained in the text.

Thresholds and Filters: The pseudocode also clarifies that no arbitrary filters (such as excluding low-citation journals) were applied. The process described was uniformly applied to every journal in our sample, ensuring methodological consistency.

Justification for Iterations: Regarding the number of iterations (1,000), this is a standard choice in computational experiments to ensure a stable and representative sampling of the distribution of possible outcomes for the randomly assembled portfolios. This allows for a robust comparison against the single "real cognitive energy" value, effectively serving as a permutation test.

We are confident that the addition of Algorithm 1 and the related clarifications have resolved the ambiguity and now provide the precise, procedural detail needed for readers to fully understand and replicate our experimental design.

Limited Generalizability

The perturbation tests focus solely on the top 500 journals by SJR, which could bias the results toward elite or multidisciplinary journals. Suggestion: Extend the analysis to a broader or stratified sample of journals across different disciplines and impact levels to test the robustness of the findings.

Response:

This question has been responsed in the previous section

Terminology Overload

Phrases like “cognitive energy landscape,” “global resampling,” and “satisficing optimization” are used without clear definition. While metaphorically appealing, they require clarification to avoid confusion. Suggestion: Define key terms clearly when first introduced and distinguish between technical concepts and metaphorical framing.

Response:

We thank the reviewer for this valuable feedback. We agree that these terms required clearer definition. We have revised the manuscript to address this, ensuring that such concepts are explicitly defined when first introduced. Furthermore, we have taken care to reduce the use of overly metaphorical language, particularly in the Results section, to ensure our empirical findings are presented as clearly and directly as possible.

Discussion

Novelty Needs Tighter Framing:

While the notion of "Mediating Similarity" is original, the discussion could benefit from tighter comparisons to alternative hypotheses (e.g., random citation behavior, reputation bias).

The implication that journals behave like agents minimizing cognitive energy is metaphorically rich, but the paper does not clarify whether this is a descriptive pattern or an agent-based modeling claim.

Response:

We thank the reviewer for this insightful feedback, which highlights two crucial areas for clarification. We have thoroughly revised the Discussion section (lines 375-456) to address both points directly.

Comparison with Alternative Hypotheses: We agree that positioning "Mediating Similarity" relative to existing theories is essential. We have expanded the discussion to explicitly contrast our principle with the classical normative and social constructivist theories of citation. Our revised argument (lines 392-430) is that "Mediating Similarity" is not a competing theory but rather a more fundamental, underlying mechanism. We propose that behaviors driven by normative rules (acknowledging intellectual debt) or social persuasion (citing authoritative, high-reputation papers) are successful heuristics precisely because they are effective strategies for navigating the "cognitive energy landscape" we describe. In this view, our principle provides a quantitative, information-theoretic foundation upon which these sociological behaviors are built.

Clarification of the "Journal as Agent" Metaphor: This is an excellent point, and we have added a dedicated paragraph (lines 403-419) to clarify our position. We now explicitly state that we are not making a literal agent-based modeling claim where journals or editors are consciously calculating KL divergence. Instead, we frame "Mediating Similarity" as a descriptive principle of an emergent, system-level phenomenon. The macroscopic pattern of energy minimization arises from the aggregation of countless microscopic, goal-oriented decisions made by individual researchers operating under bounded rationality. The "journal as an agent" is therefore an abstraction representing the aggregated citation practices of its community, and the language of physics and agency serves as a potent and predictive metaphor for this observed statistical regularity.

We believe these revisions now properly contextualize our findings, clarify the nature of our claims, and strengthen the paper's overall theoretical contribution.

Caution Needed:

Some statements verge on overclaiming, e.g., line 344–345 “...this phenomenon is not an artifact but a fundamental and universal feature.” You could qualify these claims more carefully — e.g., within the scope of the dataset used.

Response:

Thank you for this important feedback. We agree that our claims must be stated with precision and appropriately qualified by the scope of our study.

Conclusion

Restates Results Rather Than Synthesizes:

The conclusion reiterates the experiments but does not clearly articulate how this changes our understanding of scientometrics or citation behavior.

Response:

We thank the reviewer for this important point. We agree that a strong conclusion must move beyond summarizing results to articulate the broader significance and implications of the work. The original version was too focused on reiterating our findings rather than explaining their impact.

To address this, we have substantially revised the Conclusion section (lines 457-493) to focus specifically on these higher-level contributions. The new conclusion now articulates how our findings change the understanding of scientometrics and citation behavior in three key ways:

A New Theoretical Framework: We now explicitly frame "Mediating Similarity" as a new, fundamental principle that operates at a "physical" level, underlying the more commonly studied sociological motivations for citation. The conclusion now argues that our work helps resolve a "long-standing tension" by explaining how system-level coherence emerges from countless individual, boundedly rational decisions, driven by a tendency to minimize "cognitive energy."

Tangible Implications for Scientometrics: We have moved beyond simply restating results to outline three concrete, practical implications for the field:

New Evaluation Metrics: We propose that a journal's "integrative capacity"—its efficiency in bridging knowledge—can be quantified as a new dimension for assessing its role and quality, beyond traditional impact factors.

Improved Recommendation Systems: We suggest that future tools could move beyond recommending individual papers to recommending a "balanced portfolio of references" that helps authors construct a more robust cognitive bridge for their work.

A Conceptual Tool for Peer Review: We position the framework as a tool for editors, reviewers, and funding agencies to assess whether a manuscript is well-integrated into the scientific discourse or is too insular.

Future Research Directions: Finally, the conclusion now explicitly points toward new avenues for research, suggesting that the principle can be used to study the "temporal evolution of scientific fields, the emergence of disciplines, and the dynamics of innovation."

We believe the revised conclusion now effectively articulates why our findings are not just a novel observation but a significant contribution that opens new theoretical and practical avenues for the field of scientometrics.

Missed Opportunity for Broader Implications:

Consider discussing potential applications — e.g., how this understanding could be used in journal evaluation, recommender systems, or editorial policy.

Response:

This question has been responsed in the previous section

Figures and Tables

Figure 1 (Cognitive Bridge Illustration): The figure is not labeled in detail nor referenced clearly in the results. Consider improving its caption and explicitly walking the reader through it.

Figures 2–3: Ensure axes are clearly labeled (e.g., ∆1, ∆2 units), and that all visual elements (like the red dot) are described fully in the legend or text.

Response:

We thank the reviewer for their valuable suggestions to improve the clarity of our figures. We agree that more detail was needed and have acted on this feedback.

As suggested, we have redrawn the figures and provided more detailed explanations in the text.

For the "Mediating Similarity" illustration (now Figure 2), we have improved the visual labeling and expanded the caption. The main text now explicitly walks the reader through the components, defining the direct path (∆0) and the mediated path (∆1 + ∆2) to make the core concept more intuitive.

For the cognitive energy landscape plots (now Figures 3 and 4), we have redrawn them with clearer axis labels. The Y-axis is now explicitly labeled "Total Cognitive Energy," and the accompanying text (lines 288-314) now provides a thorough explanation of all visual elements: the blue curve (synthetic portfolios), the dashed line (direct path energy), and the crucial red dot (actual portfolio energy).

We are confident these revisions make the figures and our central findings much easier to interpret.

Table 1: While informative, this table would be stronger if compared with a random baseline or control.

Response:

Thanks for your advising, We added Cosine similarity as a baseline.

Writing and Language

Typographical Errors:

Line 100: “ciatation” → “citation”

Line 24: “averagely” is unidiomatic in English — consider “on average”

Redundancy: Several ideas are repeated almost verbatim across sections (e.g., lines 95–98 and 205–208). A tighter structure would enhance flow.

Tone: Generally clear and academic, though slightly more engagement with potential critiques (e.g., limits of KL divergence as a proxy for cognition) would be valuable.

Data and Code Availability

Response:

Thank you for your advise. We will conduct proof reading to ensure smooth writing

The Zenodo dataset is cited (line 390), but there is no reference to accompanying code or procedures that would enable reproduction of the perturbation experiments. If possible, release the code or explain the steps in a supplementary appendix.

Response:

Thank you for your reminder. We have shared the data first, and the code will be open source after it is sorted out.

Round 2

Reviewer 1 Report

Comments and Suggestions for Authors

Thanks a lot for all co-authors effective work. I believe that it is an interesting topic for readers. 

Author Response

Dear Reviewer:

Thank you for your professional guidance, which has helped us improve the impact and quality of our work. We express our sincere gratitude for your contributions.

Reviewer 2 Report

Comments and Suggestions for Authors

Dear Authors,

Thank you for the thoughtful and substantial revision. The manuscript is now much clearer, statistically more rigorous, and better positioned within the relevant literature. Below are concise, section-wise comments focused on this version.

Abstract

  • Nicely tightened; the core contribution and findings are clear. No further changes needed.

Introduction & Related Works

  • The intuitive explanation preceding the KL inequality and the clarified definitions of the three distributions read well.

  • The literature review is now focused and appropriately frames the system-level gap your work addresses. Good improvement.

Methods

  • The notational simplification and the toy example help a lot.

  • Please fix minor issues:

    • Equation numbering: in the distributions block, d1 and d3 are both labeled “(1)”. Please renumber sequentially.

    • Spelling: in the caption/text around Fig. 2, “cognitive brigde” → “bridge”.

  • Small robustness note (optional but helpful): you introduce additive smoothing with α=1e−9. A short sentence noting that results are insensitive to reasonable α in [1e−12, 1e−6] (or similar) would pre-empt questions about stability.

Results

  • The added explanations for the U-shape, dashed ∆0 line, and the red dot are clear and effective.

  • The inclusion of Algorithm 1 resolves the reproducibility gap for constructing candidate pools.

  • The expanded statistical treatment (percentiles as permutation logic + CIs + Cohen’s d) addresses previous rigor concerns convincingly.

  • Please fix cross-references/typos:

    • “Cognitive energy landscape vs closet journals …” → “closest” (figure captions/titles and Table 1).

    • In the perturbation paragraph, the sentence “Figure 6 shows the distribution of percentile of global perturbation. … Figure 6 shows the distribution of percentile of global perturbation.” likely intends Figure 5 (global) and Figure 6 (local). Please correct the figure numbers.

  • Optional robustness note: a one-line sensitivity check that varying the candidate-pool size (e.g., 2n vs 3n vs 4n) does not materially change conclusions would strengthen the “optimization vs. naïve proximity” claim.

Discussion

  • The positioning relative to normative and social-constructivist theories is now crisp, and the “journal as agent” clarification is appropriate. Nicely done.

Conclusion

  • The synthesis and practical implications (integrative capacity metric; portfolio-style recommendations; review heuristic) are clear and compelling.

Figures & Tables

  • Labels and captions are now informative; the added cosine baseline in Table 1 is appreciated.

  • Please correct “closet” → “closest” wherever it appears.

Data/Code Availability

  • The Zenodo data link is provided. If feasible, sharing a minimal, runnable script (even for a subset) for the perturbation pipeline would substantially enhance reproducibility. If not, a short “Reproduction Notes” appendix with commands/pseudocode and expected outputs would suffice.

Verdict

The revision satisfactorily addresses the prior major-revision points. With the minor corrections above (equation numbering, typos, figure cross-references) and optional small robustness notes, I support acceptance after minor revision.

Comments on the Quality of English Language

The manuscript reads clearly and professionally. Terminology is defined early, sentences are well structured, and the flow between sections is smooth. I consider the English fully adequate for publication, with only a few minor fixes:

  • Spelling/word choice:

    • “cognitive brigde” → “cognitive bridge” (Fig. 2 caption/text).

    • closet journals” → “closest journals” (figure captions/titles and Table 1).

  • Figure cross-references: the Results text appears to reference Figure 6 twice for global and local distributions; likely Figure 5 (global) and Figure 6 (local)—please correct.

  • Consistency: use a single form for hyphenation (e.g., “information-theoretic” throughout; choose “nonparametric” or “non-parametric” and apply consistently).

  • Very minor polishing (optional): trim a few long sentences in the Discussion for concision.

Overall, only light copyedits are needed.

Author Response

Dear Reviewer:

We are grateful for your professional guidance, which has helped us improve the impact and quality of our work. We extend our sincere respect for your contributions.

We will continue to heed your valuable suggestions, and I will now respond to each one. We have discussed why a small constant alpha does not affect the robustness of our conclusions; we have discussed why the size of the candidate pool should not exceed 3n; we have also streamlined the discussion section to make it more readable; and finally, we have uploaded the code for the perturbation experiment, so any reader can reproduce our conclusions using our open-source code and data. We have also corrected some spelling and ref errors in the article.

Thanks a lot for your professionalism and contributions.